# Accelerated Formation of Oxide Layers on Zircaloy-4 Utilizing Air Oxidation and Comparison with Water-Corroded Oxide Layers

**DOI:** 10.3390/ma16247589

**Published:** 2023-12-11

**Authors:** Shanmugam Mannan Muthu, Hyeon-Bae Lee, Bright O. Okonkwo, Dong Wang, Changheui Jang, Taehyung Na

**Affiliations:** 1Department of Nuclear and Quantum Engineering, Korea Advanced Institute of Science and Technology, 291 Daehak-ro, Yuseong-gu, Daejeon 34141, Republic of Korea; muthumech1992@kaist.ac.kr (S.M.M.); lhdory@kaist.ac.kr (H.-B.L.); or w854187495@stu.xjtu.edu.cn (D.W.); 2Institute of Corrosion Science and Technology, Guangzhou 510530, China; boo@icost.ac.cn; 3Shaanxi Key Laboratory of Advanced Nuclear Energy and Technology, State Key Laboratory of Multiphase Flow in Power Engineering, School of Nuclear Science and Technology, Xi’an Jiaotong University, Xi’an 710049, China; 4Central Research Institute, Korea Hydro & Nuclear Power Co., Ltd., Daejeon 34101, Republic of Korea; taehyung.na@khnp.co.kr

**Keywords:** Zircaloy-4, spent fuel cladding, oxide layer, corrosion, oxidation

## Abstract

For the dry storage of Canada Deuterium Uranium (CANDU) spent nuclear fuels, the integrity of Zircaloy-4 fuel cladding has to be verified. However, the formation of ~10 µm-thick oxide layers in typical CANDU reactor operating conditions takes several years, which makes sample preparation a slow process. To overcome such limitations, in this study, an accelerated formation of an oxide layer on Zircaloy-4 cladding tube was developed with a combination of high-temperature water corrosion (HT-WC) and air oxidation (AO). First, Zircaloy-4 tubes were corroded in oxygenated (2 ppm dissolved oxygen) high-temperature water (360 °C/19.5 MPa) for 500 h. Then, the tubes were air-oxidized at 500 °C for 30 h. Finally, the tubes were corroded again in HT-WC for 500 h to produce ~10 µm-thick oxide layers. The morphology and characteristics of the oxide layer in each step were analyzed using X-ray diffraction, scanning and transmission electron microscopy. The results showed that the oxide layer formed in the accelerated method was comparable to that formed in HT-WC in terms of morphology and oxide phases. Thus, the accelerated oxide formation method can be used to prepare an oxidized Zircaloy-4 cladding tube for CANDU fuel integrity analysis.

## 1. Introduction

Zirconium-based alloys have been extensively used as fuel cladding materials in water-cooled nuclear reactors like pressurized water reactors (PWR), boiling water reactors (BWR), and CANDU reactors, mostly owing to their low neutron absorption cross-section and, in part, their high strength and good corrosion resistance [1,2]. Nonetheless, during the normal operation of water reactors, the Zr-based claddings are exposed to high-temperature and high-pressure water (300–360 °C and 8–15.5 MPa), and oxide layers are formed on the surface. The thickness of the oxide layer formed on the spent nuclear fuels varies depending on water reactor types and fuel cladding materials [3,4]. For example, the oxide layer thickness could be 100–150 µm for fuel cladding of Zircaloy-4 and Nb-containing advanced Zr-based cladding in typical PWR spent fuel. Meanwhile, for CANDU reactors where Zircaloy-4 is used as a fuel cladding material, the thickness of the oxide layer is 10 µm at the time of refueling [3,4,5]. 

When removed from the reactors, spent nuclear fuels are stored in the spent nuclear fuel pool filled with highly borated water for several years until the level of decay heat drops sufficiently. Then, the spent nuclear fuels are transported to a dry storage facility, where they will be stored until permanent disposal. In the case of CANDU reactors, for dry storage, spent nuclear fuels are sealed within a container filled with inert gas, so the further corrosion of fuel cladding could be minimized during the storage period [5,6,7]. However, there is some concern that air and moisture could be trapped during the sealing process, which could cause additional corrosion of CANDU spent fuel cladding with an oxide layer, aided by the high temperature caused by the remaining decay heat from spent fuels. To assess this additional oxidation, simulated CANDU spent fuel cladding tubes with ~10 μm-thick oxide layers are needed.

## 2. Research Background

It is well known that the Zr-based cladding would be corroded in high-temperature water and form the ZrO_2_ layer. Several researchers have studied the corrosion behavior of the Zr-based cladding materials in simulated high-temperature water. Diana et al. [7] studied the long-term corrosion behavior of Zircaloy-4 in PWR primary water at 310 °C and 10 MPa for 3024 h, followed by electrochemical measurement using EIS impedance spectra. It is elucidated from the EIS plot results that the inner oxide layer acts as a barrier and the outer layer is more porous. Kim et al. [8] performed the corrosion test and analysis on the Zr-1.5Nb alloy in distilled water with a static autoclave condition at 360 °C and 18.9 MPa following the corrosion test per ASTM G2-88 [9] procedures. The authors further studied the effect of Nb-rich precipitates on the oxidation rate of the alloy. It is clearly understood from the results that the zirconium matrix oxidized faster than the β-Nb phase. Jiang et al. [10] compared the corrosion behavior of Zr-1Nb and Zircaloy-4 in LiOH-H_3_BO_3_ and pure water conditions at 330 °C and 14 MPa for 200 h. The authors observed weight loss in both LiOH-H_3_BO_3_ and pure water conditions. More weight loss was found on the Zircaloy-4 in Li/B water conditions. Also, the mechanical properties of the corroded samples were reduced as compared to the as-received samples. Kim et al. [11] performed the corrosion test on the zirconium alloy in a hydrogenated water environment at 325 °C. The oxide layer formed on the alloy surface is mainly comprised of monoclinic ZrO_2_. The authors observed the tetragonal to monoclinic ZrO_2_ phase transformation. The cracks were observed on the 200 nm-size oxide layer near the oxide–matrix interface. 

Meanwhile, studies on the oxidation of Zr-based alloys have also been performed to understand the oxidation behavior during the loss of coolant accident (LOCA) scenario, which would cause degradation of mechanical properties and the release of heat and hydrogen. When the cladding materials are exposed to air, the oxide thickness reaches a critical value, and cracks may form, which would provide the pathway for air to make direct contact with the metal surface, causing faster oxidation [12,13,14,15]. Beak et al. [16] investigated the oxidation characteristics of the Zircaloy-4 under steam conditions for 3600 s in the temperature range of 700–1200 °C. The authors observed grain coarsening, phase changes, and increasing oxide scale thickness as the temperature increased. Tung et al. [17] characterized the oxidation performance of the Zircaloy-4 cladding in an air environment at 500–800 °C for 150 h. The authors found that the oxidation rate increased with increasing the test temperature and the exposure time. Some discontinuous microcracks were noticed on the scale formed on the alloy above 650 °C. Jordan et al. [13] compared the oxidation behavior of the pure Zr, Zr-3, Zircaloy-4, Zr-1Nb, and Zr-2.5 Nb alloy samples in an air and oxygen environment from 400–800 °C. The authors reported that all the samples experienced to faster breakaway oxidation and formed a thick scale. Less protective oxide scales formed on the alloys after oxidation in air environment as compared to the oxygen. 

The main purpose of this study is to produce a 10 µm-thick oxide layer on the Zircaloy-4 cladding to simulate the CANDU spent fuel cladding condition by combining high-temperature water corrosion and air oxidation to accelerate the growth of the oxide layer. To check the validity of this approach, the characteristics of the oxide layer formed by the accelerating process were compared with those formed in high-temperature water. For the characterization, X-ray diffraction (XRD), scanning electron microscope (SEM), and transmission electron microscope (TEM) analyses were performed. The results were compared and discussed in view of the applicability of the accelerated process to produce the simulated CANDU spent fuel cladding. 

## 3. Experimental Methods

### 3.1. Materials 

Zircaloy-4 is extensively used as a fuel cladding tube in the nuclear industry. In this research work, a tube of Zircaloy-4 was chosen as the substrate material, provided by the KEPCO Nuclear Fuel Co., Ltd. (Daejeon, Republic of Korea). The elemental analysis was performed on the substrate using inductively coupled plasma optical emission spectroscopy (ICP-OES), and the compositions were Zr-1.45%Sn-0.2%Fe-0.1%Cr. 

### 3.2. High-Temperature DO Water Corrosion and the Oxidation Test 

For the corrosion test, the specimens were prepared in a cylindrical tube with dimensions of 10 mm in length and 0.4 mm in thickness, and the outer diameter of the tube was 10 mm. Prior to the test, the specimens were polished with SiC sheets with 600 grit size followed by rinsing with acetone to remove the oxides and other impurities. The average surface roughness of the substrate material is 0.12 µm. The high-temperature water corrosion (HT-WC) test in a oxygenated HTW environment and the air oxidation (AO) test were carried out on Zircaloy-4 in three different conditions and the study diagram is shown in Figure 1. These were the following:

Case I: Corrosion in a oxygenated high-temperature water (HT-WC) for 4000 h.

Prior to the corrosion test, the samples were hanged in the autoclave using stainless steel wires. The high-temperature water corrosion (HT-WC) test was conducted on the Zircaloy-4 samples in a simulated primary water reactor environment containing lithium hydroxide (LiOH-2.2 ppm, Junsei Chemical Co., Ltd., Tokyo, Japan) and boric acid (H_3_BO_3_-1200 ppm, Daejung Chemicals and Metals Co., Ltd., Siheung, Republic of Korea) at a temperature of 360 °C and 19.5 MPa. To accelerate the oxidation rate of Zircaloy-4, a dissolved oxygen (DO) content of 2000 ppb was used. The test was performed up to 4000 h with an interval of 24, 100, 300, 500, 1000, 2000, 3000, and 4000 h. The specimen weight was measured at the end of each exposure time to determine the oxidation rate by the weight change method. Initially, 24 specimens were loaded, then 3 specimens were removed at each interval and measured, and the remaining specimens were further exposed till next interval. 

Case II: HT-WC for 500 h and air oxidation (AO) in an air environment at 500 °C for 30 h.

In Case II, the samples were subjected to HT-WC for 500 h, followed by air oxidation at 500 °C (AO) in an air environment for 30 h. The weight change was monitored regularly at 10, 20, and 30 h. Initially, 10 specimens were exposed to HT-WC and subjected to AO. One specimen was removed after 10 and 20 h AO. All remaining 8 specimens were exposed to 30 h AO, of which 3 were used for measurement and analysis. The remaining 5 specimens were used for Case III test.

Case III: HT-WC for 500 h, AO at 500 °C for 30 h, and HT-WC for 500 h. For this test, 5 specimens from Case II were further exposed to HT-WC for 500 h. In Case III, the samples were subjected to HT-WC for 500 h and AO at 500 °C for 30 h. Then, the samples were again exposed to high-temperature water for 500 h. 

### 3.3. Analysis Methods 

The change in weight of the samples was measured using an electronic weighing balance (Toledo, OH, USA), with an accuracy of 0.0001 g. The surface morphology and elemental analysis of the oxide scale was characterized by a scanning electron microscope (FE-SEM: Hitachi SU-5000, Hitachi Hi-Tech, Tokyo, Japan) equipped with energy dispersive spectroscopy (EDS). The phase analysis of the oxide layer was performed by X-ray diffraction analysis (XRD: RIGAKU D/MAX-2500, RIGAKU Corp, Tokyo, Japan) ranging from 20° to 90° with a step size of 0.01° and a scan speed of 4° min^−1^. To determine the thickness of the oxide layer, cross-sectional analysis was performed on the corroded samples using a Focused Ion Beam (DB-FIB: FEI Helios G4 UX, Thermo Fisher Scientific solutions Co., Ltd., Seoul, Republic of Korea) and a transmission electron microscope (TEM: FEI Talos F200X, Thermo Fisher Scientific solutions Co., Ltd., Republic of Korea). The FIB milling process was employed to prepare the TEM sampling, and the sputtering was obtained with a Ga^+^ ion beam. The elemental distribution on the oxide layer was examined using EDS mapping and line scan analysis. High-resolution TEM (HR-TEM) images and the Fast Fourier Transform (FFT) diffraction (SAED) pattern were obtained using HR-TEM images.

## 4. Results

### 4.1. Weight Gain Measurement 

The weight change in the samples per unit area was calculated using the following Formula (1) [18].
(1)$$∆W=\frac{{W_{f}-W}_{i}}{A}$$
where Δ*W* is the weight change in the samples per unit area; *W_i_* and *W_f_* are the initial and final weight of the samples and *A* is the surface area of the samples. The weight gain measurement results are shown in Figure 1. For Case I (Figure 2a), the weight gain of the samples increased rapidly until 500 h, reaching 25.24 mg/dm^2^ with parabolic behavior. During the initial stage, the weight gain of the sample is high due to the formation of a new oxide layer on the surface. Then, the rate slowed down until 2000 h, followed by the linear weight gain after 2000 h. This may be attributed to the formation of cracks, resulting in oxide growth rate was increased slightly. Finally, the weight gain of the samples was 64.17 mg/dm^2^ after being exposed for 4000 h for Case I. 

In Case II, the sample was initially subjected to HT-WC for 500 h, followed by air oxidation at 500 °C for 30 h. Figure 2b illustrates that the weight gain of the sample was 125.10 mg/dm^2^. In Case III, the samples were exposed to 500 h in the HT-WC condition, 30 h in air oxidation, and again 500 h in the HT-WC condition. The weight gain of sample was 153.97 mg/dm^2^. It is clear from the weight gain plot that the samples experienced a larger weight gain during the air oxidation test. The weight gain results of Zircaloy-4 after corrosion in three different conditions are listed in Table 1.

### 4.2. XRD Analysis

To determine the oxide phases, XRD analysis was performed on the bare and corrosion tested Zircaloy-4 samples and the results are shown in Figure 3. Zr phase was found on the bare Zircaloy-4 sample. There are no oxide phases were observed on the bare sample. For all cases, strong peaks of Zr matrix (PDF #05-0665) and m-ZrO_2_ (PDF #37-1484) were observed. In addition, a very small peak (at 30.1°) of t-ZrO_2_ (PDF #50-1089) was observed. The relative intensities of m-ZrO_2_ peaks were greater for Case III, which is consistent with the higher weight gain of Case III. Generally, t-ZrO_2_ would appear when the zirconium-based alloys are exposed to temperatures above 1070 °C; however, it could be formed under high stress of the oxide scale [19,20]. 

### 4.3. Characterization of the Oxide Layer

#### 4.3.1. Case I: HT-WC for 4000 h

The surface morphology of the oxide layer developed on the sample after exposure to a HT-WC for 4000 h is displayed in Figure 4a,b. It can be seen from the SEM images with different magnifications that the oxide scales covered the entire surface of the sample. Along with the fine particle shapes, nodule oxides are found on the scales, which are spread on the entire sample surface. The XRD results (Figure 3) suggest that the oxide scales are ZrO_2_. The cross-section SEM image of the Zircaloy-4 after HT-WC for 4000 h is displayed in Figure 4c. The average thickness of the oxide layer was 5.12 µm for the outer sides of the Zircaloy-4 tubes. For detailed oxide scale analysis, the TEM sampling was prepared using FIB equipped with SEM. The SEM image of FIB fabricated area of the corroded Zircaloy-4 sample after HT-WC for 4000 h is shown in Figure 4d. Some microcracks, pores, and a large bubble were found formed on the scale of Zircaloy-4 after subjected to HT-WC for 4000 h.

Figure 5 shows the results of a TEM-EDS cross-sectional analysis results of FIB sample of the oxide layer that formed on Zircaloy-4 exposed to HT-WC for 4000 h. It can be seen clearly from the TEM image that there is a bubble in the top part of the ZrO_2_ oxide layer. In addition, several horizontal cracks were noticed on the scale. The EDS mapping results show that bubble is formed between the inner ZrO_2_ and outer Fe-rich oxides. A vertical crack was present at the top surface of the ZrO_2_ scale in the bubble, which is clearly highlighted. The formation of bubbles has been well reported in the literature [20,21,22]. The main reasons behind the bubble formation on the top region of the oxide layer are (i) the large variation in the thermal expansion co-efficient between the ZrO_2_ matrix and the Fe-rich oxide in the water; and (ii) longer exposure time of the oxide layer in a high-temperature water medium. 

The EDS line scanning analysis was carried out on the oxide scale to study the elemental distribution. The results of the EDS line scan were in good agreement with the EDS mapping results, showing the presence of outer Fe-rich oxide and inner ZrO_2_. The HR-TEM and FFT pattern results (Figure 6) illustrate that the oxide scale present above the bubble region is mainly composed of Zr, Fe, and O-rich phases. The FFT pattern of region i indicates the presence of the hexagonal closed pack (hcp-ZrFe_2_) phase, which is in agreement with the literature [23,24]. Meanwhile, in region ii, near a crack, the amorphous m-ZrO_2_ was identified.

The formation of Fe-rich oxide and ZrFe_2_ phase on Zircaloy-4 (Fe content of 0.2%) during exposed HT-WC conditions could be attributed to the diffusion of Fe from the autoclave made of stainless steel. Similarly, Jeong et al. [25] found the Zr_3_Fe precipitate after the corrosion test on the Zr-Nb alloy in high-temperature water conditions at 360 °C and supercritical water conditions at 500 °C.

#### 4.3.2. Case II: HT-WC for 500 h + AO for 30 h

The surface morphology of the oxide layer developed on the sample after exposure to a HT-WC for 500 h followed by AO for 30 h (Case II) is displayed in Figure 7a,b. It is clear that, compared to the Case I sample (Figure 4), more and larger oxide nodules were found on the air-oxidized surface of the Case II sample. Figure 7c,d illustrates the SEM and FIB-EDS results of the oxide layer of the Zircaloy-4 exposed to HT-WC for 500 h followed by AO for 30 h. FIB image clearly show that the oxide layer contains several cracks, horizontal to the interface of oxide and metal. Ishii et al. [26] observed the cracks on the oxide scales during the oxidation studies on the Zircaloy-2 samples in air environment at 450 °C. The main reason for the formation of cracks in the oxide scale is generation of compressive stresses during the oxidation [26,27]. The scale thickness was measured ~10 µm. The EDS line scanning result (Figure 7e) shows the scale mainly consisted of Zr and O. The O content was very low (10%) in the transition region, which is found between the scale and matrix regions. 

TEM and EDS mapping analysis were performed on the oxide layer formed on the sample exposed to a HT-WC for 500 h followed by AO for 30 h (Case II) at different regions, such as the top region of the oxide layer and the oxide–matrix interface. TEM images (Figure 8) clearly display the presence of several micro- and nano-sized cracks at the top region of oxide layers. These kinds of cracks were not identified by the SEM analysis. The scales are mainly rich in Zr, O, and Sn. In addition, a Fe-rich oxide precipitate was observed in the top region of the scale, which was formed during exposure to high-temperature water prior to air oxidation. TEM analysis at the oxide–matrix interface region is shown in Figure 9. The cracks were found in the oxide layer close to the oxide–matrix interface. The transition region is clearly visible by TEM analysis, and the thickness of the transition region is ~200 nm. In the transition region, the oxygen was partially distributed, which suggests that the transition region was formed by the diffusion of oxygen into the matrix [28,29]. HR-TEM analysis was carried out at the top and oxide–matrix regions of oxide layers and the results are shown in Figure 10a,b. The FFT pattern result demonstrates that the oxide layer is mainly composed of m-ZrO_2_. 

#### 4.3.3. Case III: HT-WC for 500 h + AO for 30 h + HT-WC for 500 h 

The surface morphology of the oxide layer developed on the sample exposed to a HT-WC for 500 h, followed by AO for 30 h, and again HT-WC for 500h (Case III) is shown in Figure 11. Several small and nodular oxides were evenly distributed on the entire surface of the oxide scales. In addition, some microcracks were clearly seen on the oxide layer. The overall surface morphology and distribution of nodule oxides are similar to those for Case I and Case II. The cross-sectional SEM image (Figure 11c) shows that the average scale thickness is ~12.8 µm and the EDS line scanning result (Figure 11d) indicates that the scale is ZrO_2_. The FIB image (Figure 11e) reveals the existence of several larger cracks in the oxide layer.

TEM and EDS mapping analysis were performed on the oxide layer formed on the sample exposed to a HT-WC for 500 h followed by AO for 30 h (AO) and HT-WC for 500 h (Case III) at different regions, such as the top region of the oxide layer, and the oxide–matrix interface. The TEM image (Figure 12) shows that the top region of the oxide layer contains the mixture of equiaxed and columnar grains. Similar to Case I, a large bubble was present at the top region of the oxide layer, but the presence of Fe-rich oxide was not clear above the bubble. The EDS mapping results show that Zr and O were mainly distributed in the top region of the oxide layer while Fe content was almost negligible. Cracks were not found in the observed region, but large cracks appeared somewhat below the top region (~5 µm from top surface). The results of TEM-EDS analysis at the oxide–matrix interface are shown in Figure 13. The cracks were found in the oxide layer close to the oxide–matrix interface. The transition region observed for Case II sample was not clear for Case III sample. HR-TEM analysis was carried out at the top and oxide–matrix regions of the oxide layer and the results are shown in Figure 14a,b. The FFT pattern result demonstrates that the oxide layer is mainly composed of m-ZrO_2_. It should be noted that, though Fe-rich oxides and intermetallics were not observed in the region where TEM analysis was performed, their presence could be anticipated due to the similarity in corrosion environment to Case I. In addition, the presence of hydrides was observed at the oxide–matrix interface. 

## 5. Discussion 

### 5.1. Oxide Layer Formation on the Zircaloy-4 in HT-WC Environment 

In this study, the weight gain (mg/dm^2^) was found on the Zircaloy-4 after being exposed to a HT-WC environment for 4000 h due to the growth of a new oxide layer. The sample weight was increased with respect to exposure time. The XRD results confirmed the Zr-matrix and m-ZrO_2_ phases present on the oxide scale. The formation of the ZrO_2_ oxide phase on Zicaloy-4 is due to the diffusion of oxygen ions into the Zr matrix during exposure to a water medium at 360 °C. The oxide layer formation mechanism was clearly explained by the following Equation (2).
(2)$${Zr}^{4+}+{2O}^{2-}\to {ZrO}_{2}$$

Ni et al. [28] observed the distribution of ZrO_2_ at the oxide–matrix interface of PWR-corroded Zircaloy-4 at 330 °C. But the ZrO_2_ layer is not continuous due to the short immersion time and low water temperature. Along with this, transparent layer grains were developed with a size of 30–500 nm. It is well known that the corrosion behavior of zirconium materials depends mainly on the oxide characteristics, such as grain morphology and crystal structure [25]. In addition, other elements present in the zirconium alloys also lead to the formation of oxide precipitates, which also affect the oxidation properties. From the XRD and EDS results, no other precipitates were found on the ZrO_2_ layer due to the occurrence of low Sn (1.45%) in the Zircaloy-4. But Hong et al. [30] found that increasing the amount of Sn in Zircaloy-4 made it less resistant to oxidation in the PWR at 360 °C and 180 bars. Jeong et al. [31] investigated the influence of Sn addition with Zr on the oxidation performance of the Zr-alloy in LiOH conditions. The authors concluded that the addition of Sn content (1%) to the Zr alloy would resist Li penetration into the oxide scale and further reduce the oxidation rate.

Based on the crystal structure, ZrO_2_ can be classified as monoclinic and tetragonal. According to the grain shape, it is divided into columnar and equiaxed crystals. It is well known that the existence of columnar and tetragonal phases on the oxide layer could enhance the oxidation resistance. In this current study, Zr-matrix and m-ZrO_2_ phases are the predominant oxide phases. Generally, the t-ZrO_2_ phase will appear above the oxidation temperature of 1070 °C. However, a minor amount of t-ZrO_2_ phase was observed due to high compressive stress, which could stabilize the tetragonal phase. At the initial stage, a tetragonal phase was formed, which was changed into a monoclinic When the Zircaloy-4 is subjected to oxidation, volume expansion may occur on the scale, resulting in compressive stresses. During the relaxation of the compressive stress, the tetragonal oxide phase destabilized and transformed into the monoclinic phase. Due to stress reduction in the oxide scale, the tetragonal phase was changed to the monoclinic phase, which leads to crack development. 

### 5.2. The Bubble Formation Mechanism 

The FIB image (Figure 4d) clearly indicates the existence of bubbles on the top surface of the oxide scales that formed Zircaloy-4 after 3000 h and 4000 h of exposure to HT-WC. The bubble formation may cause the oxide layer to fail during HT-WC at elevated temperatures. Therefore, it is significant to study the bubble formation mechanism. For the detailed discussion, the images with higher magnification were obtained from the FIB images of the HT-WC sample. The length of the bubble was found to be in the range of 3–4 µm. 

Three different stages were involved in the bubble formation process. These are the initial bubble formation stage, the bubble accumulation stage, and the bubble connection stage. Generally, the Kirkendall mechanism was used to analyze the initial bubble formation mechanism, which has been stated by several researchers during the oxidization test on the Zircaloy-4. The main reasons behind the bubble formation on the top region of the oxide layer are (i) the large variation in the thermal expansion coefficient between the ZrO_2_ matrix and the Fe-rich oxide in the water medium; and (ii) rapid oxide growth by oxygen diffusion resulting in compressive stress. Therefore, several cracks were formed on the scales, which affect the oxidation properties at high pressure and temperature [21,32,33]. 

### 5.3. Air Oxidation Mechanism

The air oxidation test was conducted on the HTW-corroded Zircaloy-4 samples at 500 °C for 30 h to study the oxidation characteristics of the HTW-corroded surface. It can be noticed from the weight gain plot that the oxidation rate is high in the air oxidation (AO) condition as compared to the HT-WC condition. This may be attributed to the higher test temperature (500 °C) and large amount of oxygen in the air oxidation test as compared to the HT-WC test. The ZrO_2_ oxides were developed on the Zircaloy-4 surface due to the reaction of oxygen ions from the atmospheric air and the Zircaloy-4 matrix. The oxide growth occurred continuously due to the penetration of oxygen through the oxide layer. Therefore, the volume of the oxides formed from the metal is higher than the metal (the Pilling–Bedworth ratio (PBR) of zirconium and zirconium oxide is 1.56) [29,34]. 

It is essential to consider that several factors, such as morphology, crystal structure, porosity, and cracks, are present on the scales when examining the oxidation characteristics of the Zr-based alloys. The surface morphology of the oxidized surface (Figure 7) illustrated more oxide growth on the surface than on the HTW-corroded surfaces. Along with that, more nodule-rich oxide phases were noticed. When the HTW-corroded Zircaloy-4 during oxidation in air, a compressive stress was induced on the oxide scale, causing a tetragonal phase was developed at the initial stage. During the cooling period in the atmospheric air, outward oxide growth occurred on the corroded surface, causing the compressive stress to be relaxed. Therefore, the tetragonal phase was changed to the monoclinic phase. The monoclinic phase is less protective as compared to the tetragonal phase. A thick scale was formed on the HTW-corroded surface after air oxidation. Several cracks were developed on the scale, which are parallel to the metal–oxide interface. These cracks were mainly attributed to the cyclic growth of stress on the oxide scale during heating and cooling. In addition, the Pilling–Bedworth ratio (PBR) of ZrO_2_/Zr is 1.56 because high-volume expansion may induce more stress on the scale [24,27]. TEM results show the existence of several cracks on the scale of the oxidized sample, which are not able to be detected by SEM cross-section analysis. Moreover, Fe- and O-rich precipitate were found on the top surface of the scale, and Sn was partially distributed on the scale. The oxide–matrix interface region was analyzed by TEM to study the adhesion behavior. The thickness of the scale was found in the range of 1.3–1.5 µm after HT-WC for 500 h. Cracks are clearly seen by TEM. These cracks may increase the oxide growth rate. This may also be one of the reasons for higher oxidation when exposed to air. The white region was noticed in the matrix region below the oxide scale for 200 nm. This region is called the oxygen transition region, which indicates that the oxygen was distributed into the Zr matrix region. 

### 5.4. Comparison of Oxide Layer Characteristics 

The characteristics of oxide layers formed in three cases of corrosion and oxidation conditions are summarized in Table 2. It can be said that the characteristics of oxide layers formed in Case I and III differ only in terms of thickness and the presence of bubbles on the top region of the oxide layer. However, it is anticipated that the presence of bubbles would not affect the overall integrity of the oxide layer during long-term storage in a dry storage facility because of the location of bubbles and the relatively smaller size compared to oxide layer thickness. 

## 6. Conclusions 

In this work, the Zircaloy-4 fuel cladding tubes were subjected to a combination of corrosion in oxygenated high-temperature water at 360 °C/19.5 MPa (HT-WC) and air oxidation at 500 °C (AO) to form a ~10 µm-thick oxide layer on the surface. The following three cases of corrosion and oxidation conditions were used and the resulting oxide layers were compared: 

Case I: HT-WC for 4000 h,

Case II: HT-WC for 500 h + AO for 30 h, and

Case III: HT-WC for 500 h + AO at 500 °C for 30 h + HT-WC for 500 h. 

Based on the tests and analysis, the following conclusions were drawn:The weight gain was 64.17 mg/dm^2^ after HT-WC for 4000 h (Case I) and the scale thickness was found to be ~5 µm. Meanwhile, the weight gain was 153.97 mg/dm^2^ after HT-WC for 500 h followed by AO for 30 h and HT-WC for 500 h (Case I) and the scale thickness was ~13 µm. Thus, air oxidation showed accelerated weight gain and oxide layer growth compared to high-temperature water corrosion.In all three cases, the oxide layer mainly consisted of m-ZrO_2_ with little t-ZrO_2_ confirmed by the XRD analysis and TEM analysis. Also, horizontal cracks were widely present from the top of the oxide layer to the oxide–matrix interface.The oxide layer formed in the accelerated method (Case III) was comparable to that formed in HT-WC in terms of morphology and oxide phases. Thus, the accelerated oxide formation method can be used to prepare an oxidized Zircaloy-4 cladding tube for CANDU fuel integrity analysis.

It should be noted that, even though the accelerated formation of the oxide layer by adopting air oxidation was successful, the microstructure change and resulting mechanical property change in Zr matrix should be investigated in the future.

## Figures and Tables

**Figure 1 materials-16-07589-f001:**
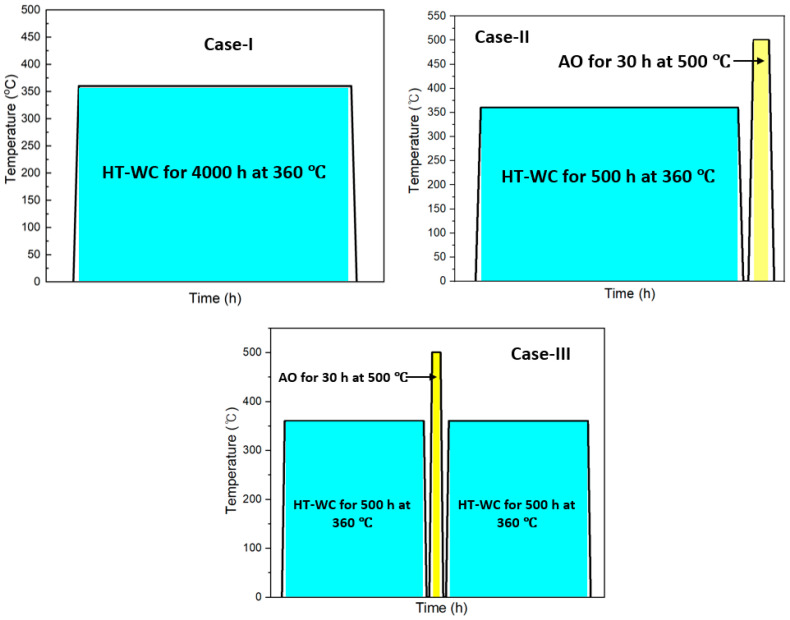
The study diagram of the Zircaloy-4 after subjected to different cases.

**Figure 2 materials-16-07589-f002:**
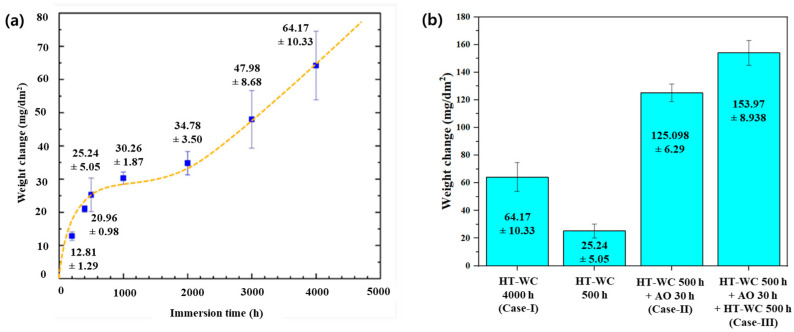
Weight gain plot of the Zircaloy-4 after exposure to (**a**) high-temperature water corrosion (HT-WC) condition and (**b**) a combination of HT-WC and air oxidation (AO) conditions.

**Figure 3 materials-16-07589-f003:**
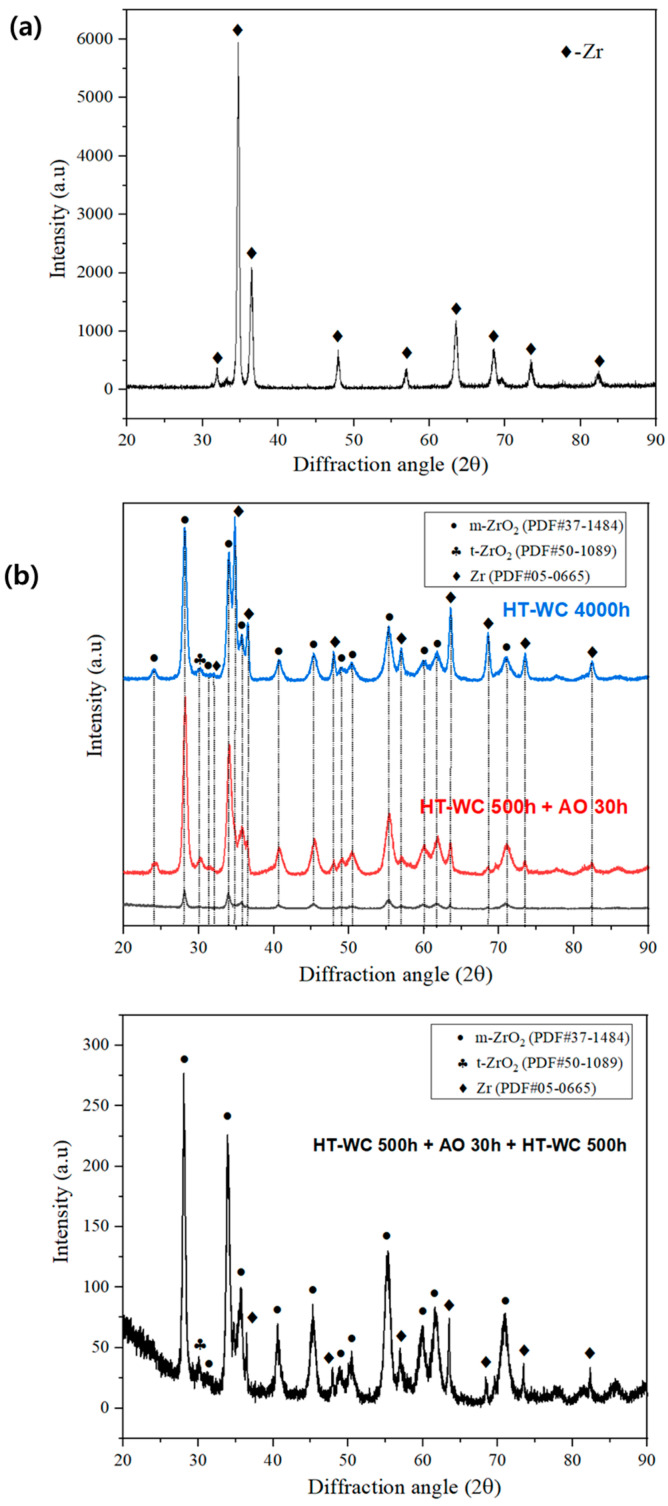
XRD analysis of the (**a**) bare and (**b**) corroded Zircaloy-4 after exposure to high-temperature water corrosion (HT-WC) and an air oxidation environment.

**Figure 4 materials-16-07589-f004:**
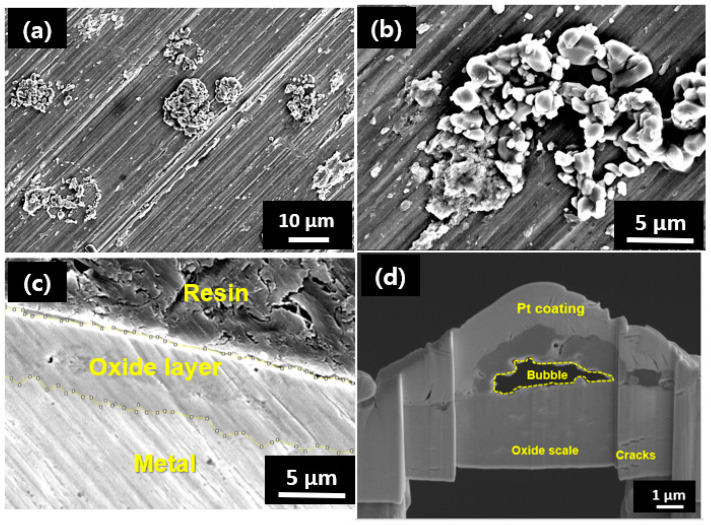
SEM (**a**,**b**) surface, (**c**) cross-sectional images, and (**d**) FIB image for the TEM sample of Zircaloy-4 exposed to high-temperature water (HT-WC) for 4000 h (Case I).

**Figure 5 materials-16-07589-f005:**
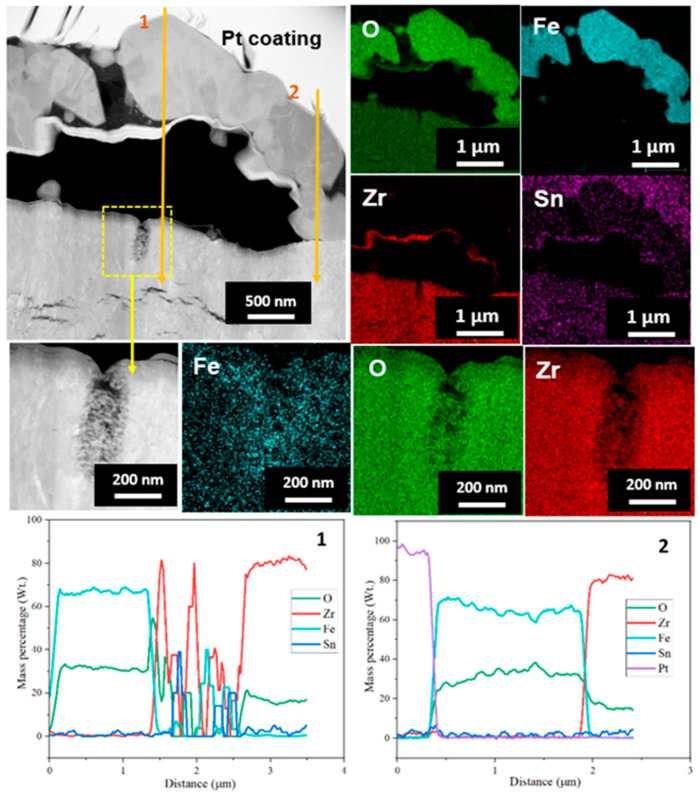
TEM image and EDS analysis results of the oxide layer on Zircaloy-4 exposed to the HT-WC for 4000 h (Case I).

**Figure 6 materials-16-07589-f006:**
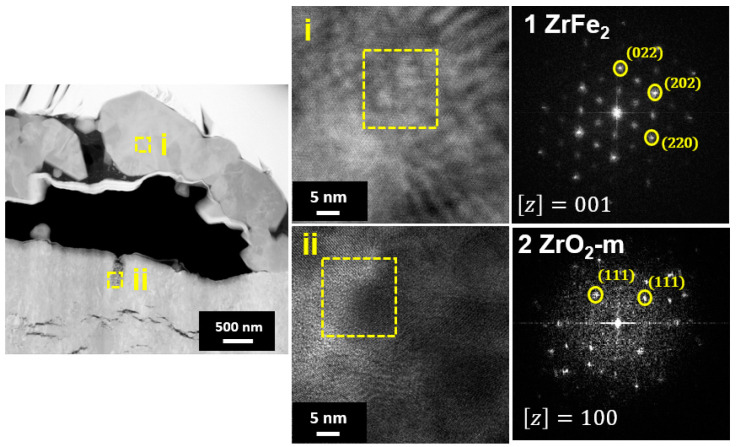
HR-TEM micrograph and FFT patterns of the oxide layer on the Zircaloy-4 exposed to the HT-WC for 4000 h (Case I).

**Figure 7 materials-16-07589-f007:**
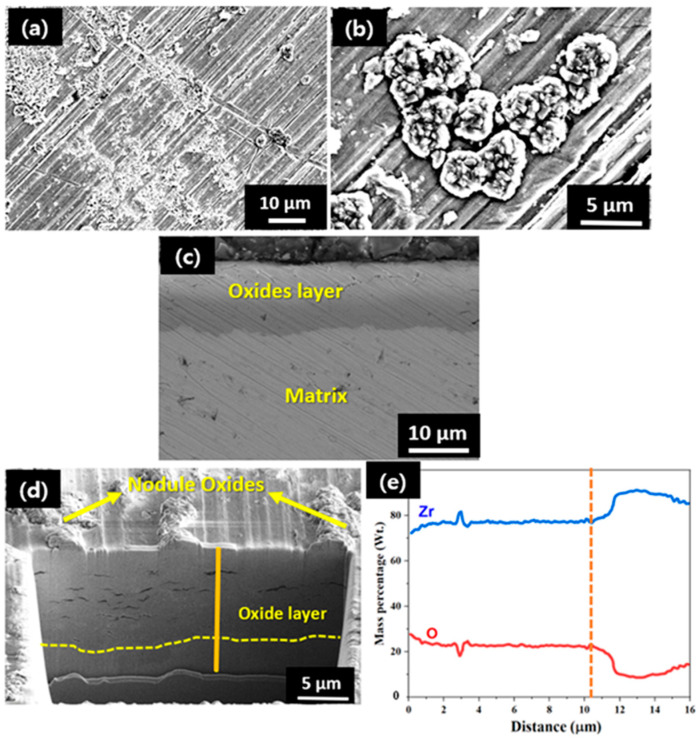
(**a**–**c**) SEM analysis at the surface and cross-section and (**d**,**e**) FIB-EDS analysis of the Zircaloy-4 exposed to HT-WC for 500 h followed by AO for 30 h (Case II).

**Figure 8 materials-16-07589-f008:**
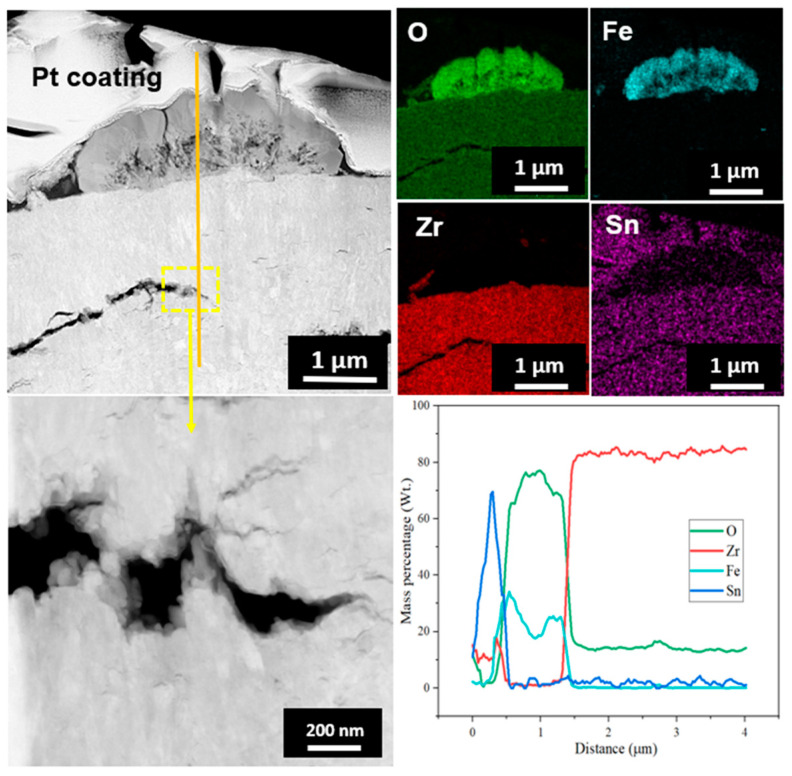
TEM-EDS analysis results at the top region of the oxide layer of Zircaloy-4 exposed to HT-WC for 500 h followed by AO for 30 h (Case II).

**Figure 9 materials-16-07589-f009:**
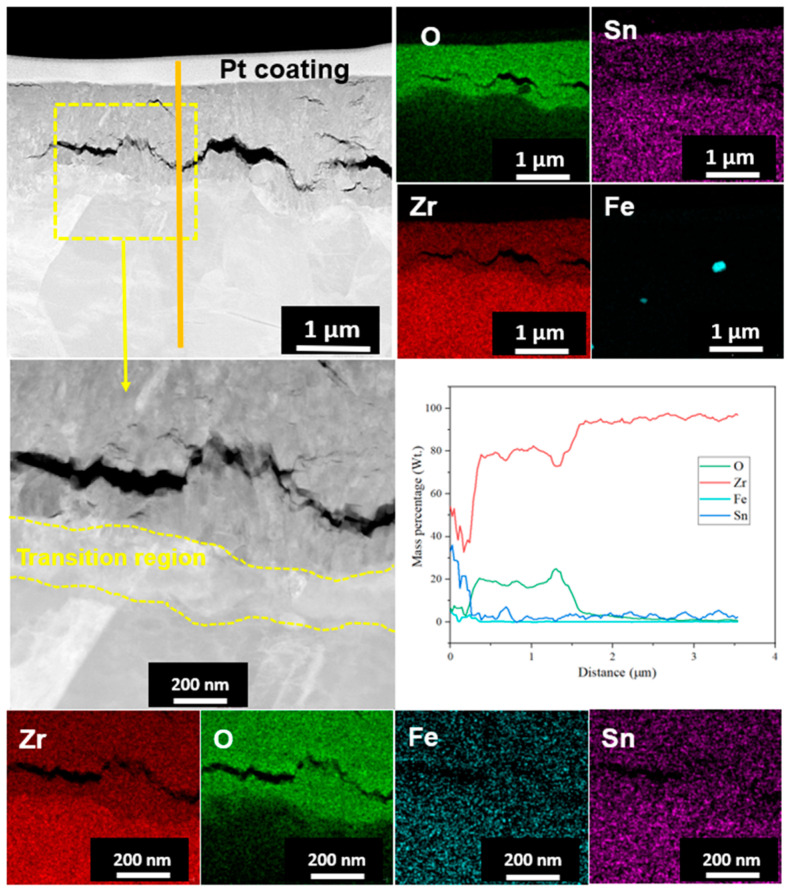
TEM-EDS analysis results at the metal–oxide interface region of the oxide layer of Zircaloy-4 exposed to HT-WC for 500 h followed by AO for 30 h (Case II).

**Figure 10 materials-16-07589-f010:**
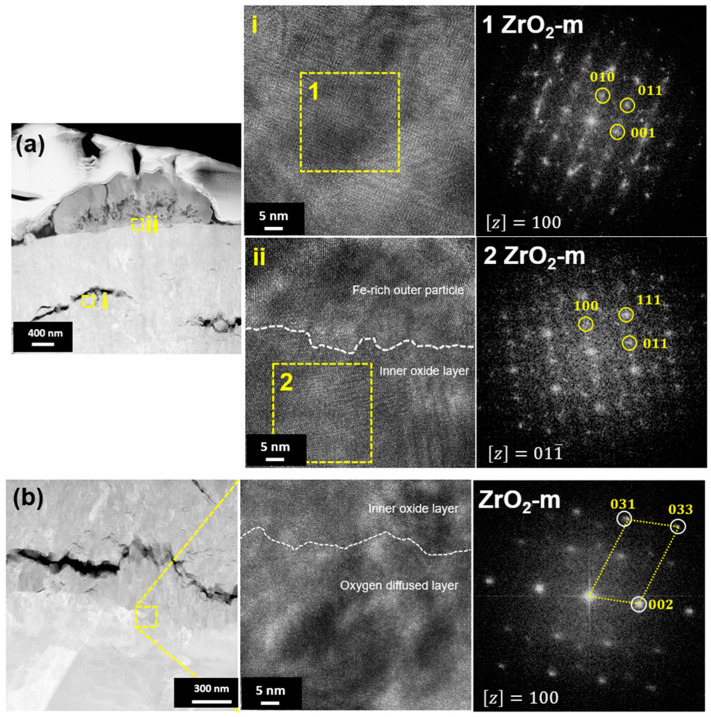
HR-TEM micrographs and FFT patterns at the (**a**) top and (**b**) metal–oxide interface regions of the oxide layer of Zircaloy-4 exposed to HT-WC for 500 h followed by AO for 30 h (Case II).

**Figure 11 materials-16-07589-f011:**
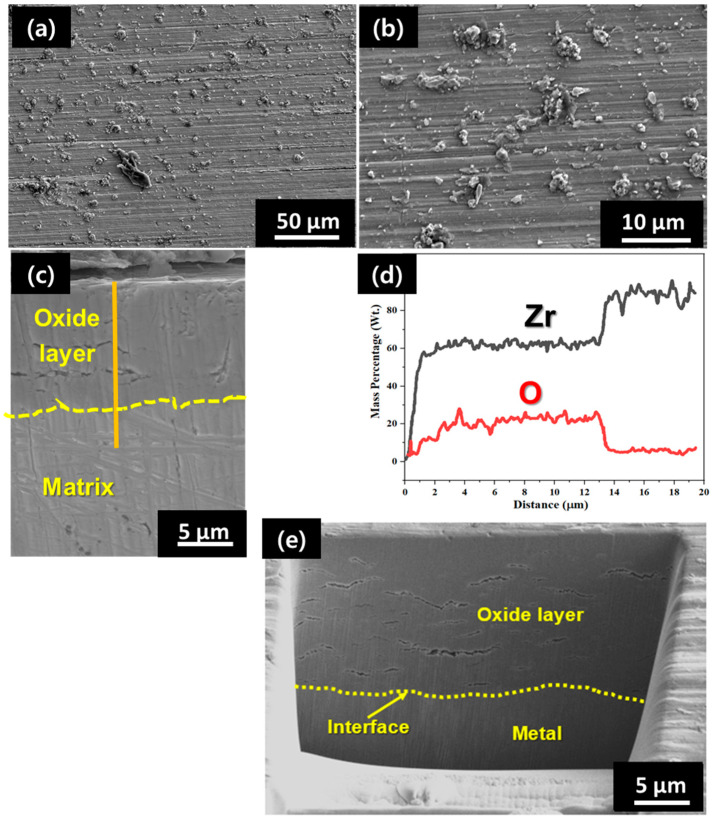
(**a**,**b**) SEM-EDS at the surface and (**c**,**d**) cross-section and (**e**) FIB analysis of the oxide layer of Zircaloy-4 exposed to HT-WC for 500 h followed by AO for 30 h and HT-WC for 500 h (Case III).

**Figure 12 materials-16-07589-f012:**
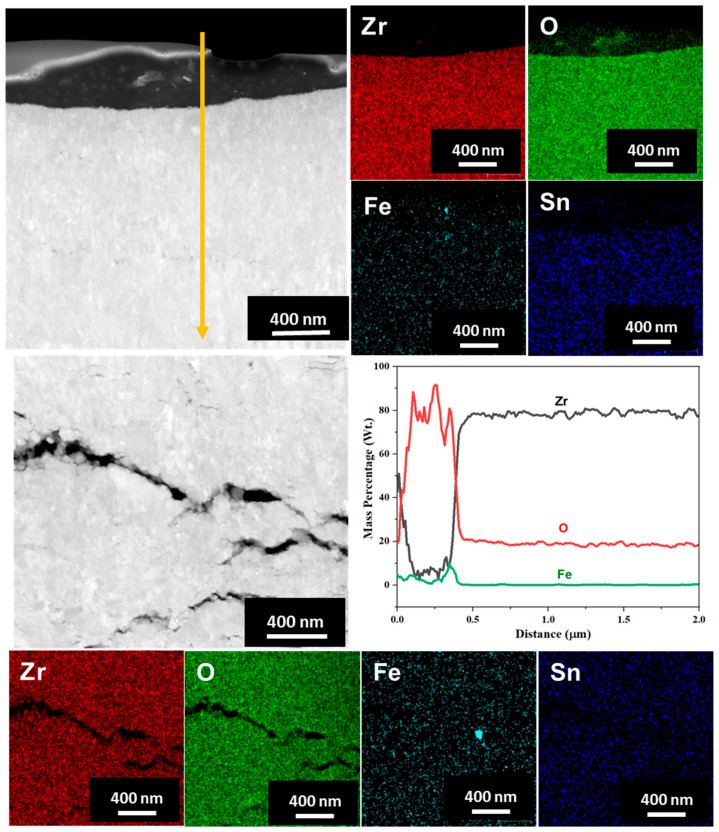
TEM and EDS mapping results at the top region of the oxide layer formed on the Zircaloy-4 exposed to HT-WC for 500 h followed by AO for 30 h and HT-WC for 500 h (Case III).

**Figure 13 materials-16-07589-f013:**
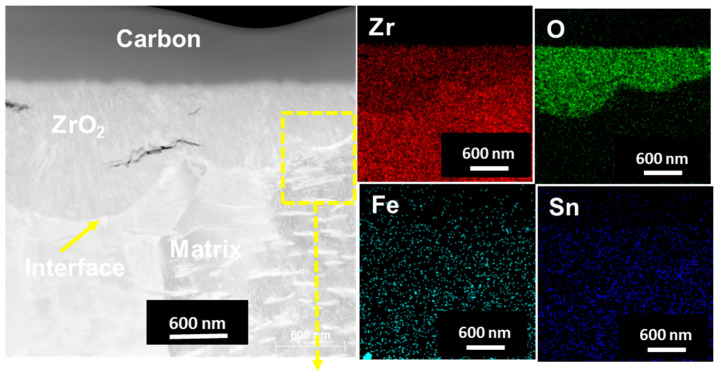
TEM and EDS mapping results at the oxide–matrix interface of the Zircaloy-4 exposed to HT-WC for 500 h followed by AO for 30 h and HT-WC for 500 h (Case III).

**Figure 14 materials-16-07589-f014:**
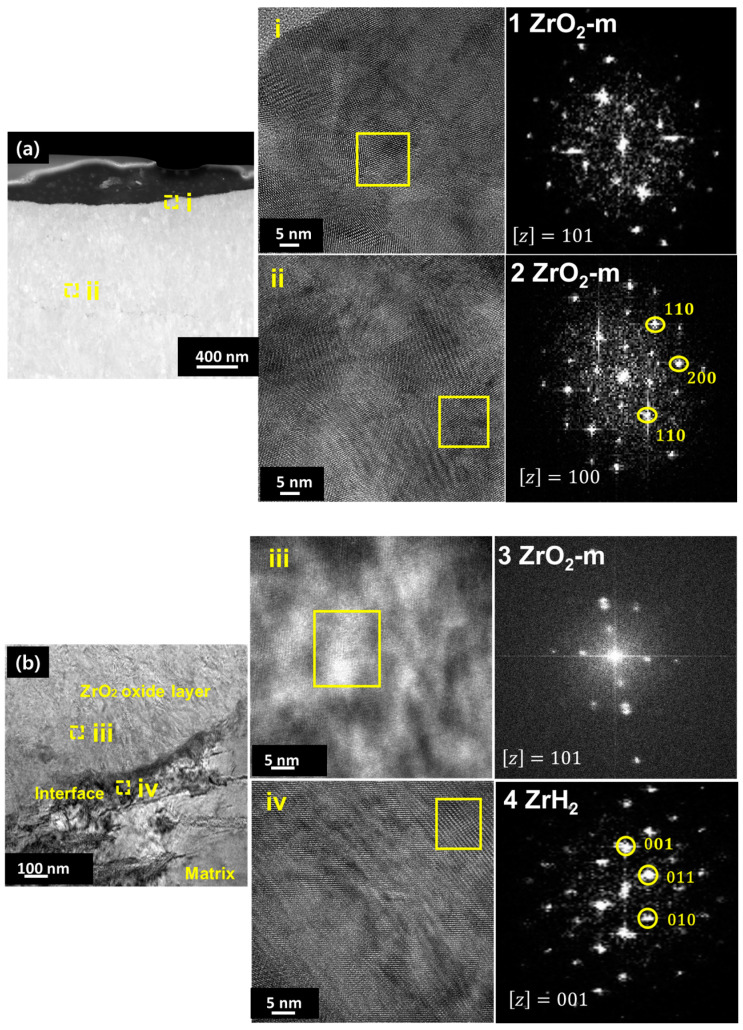
HR-TEM micrographs and FFT patterns at the (**a**) top and (**b**) metal–oxide interface regions of the oxide layer of Zircaloy-4 exposed to HT-WC for 500 h followed by AO for 30 h and HT-WC for 500 h (Case II).

**Table 1 materials-16-07589-t001:** Weight gain of the Zircaloy-4 after exposure to different conditions.

Cases	Condition	Weight Gain (mg/dm^2^)
I	HT-WC for 4000 h	64.17
II	HT-WC for 500 h + AO for 30 h	125.10
III	HT-WC for 500 h + AO for 30 h + HT-WC for 500 h	153.97

**Table 2 materials-16-07589-t002:** Comparison of the characteristics of the oxide layer formed on Zircaloy-4 exposed to three cases of corrosion and oxidation conditions.

Cases	Oxide Thickness	Oxide Phase	Location of CRACKS	Bubbles
I	~5 μm	Mostly m-ZrO_2_ + little t-ZrO_2_	Top layer to interface	Yes
II	~10 μm	Mostly m-ZrO_2_ + little t-ZrO_2_	Top layer to interface	No
III	~13 μm	Mostly m-ZrO_2_ + little t-ZrO_2_	Top layer to interface	No

## Data Availability

The raw/processed data required to reproduce these findings cannot be shared at this time due to technical or time limitations. They will be shared upon request.

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
