# Peer review of "Accelerated Formation of Oxide Layers on Zircaloy-4 Utilizing Air Oxidation and Comparison with Water-Corroded Oxide Layers"

_materials, 2023, doi:10.3390/ma16247589_

Round 1

Reviewer 1 Report

Comments and Suggestions for Authors

The article provides an investigation into the weight change and oxide layer characteristics of Zircaloy-4 under different corrosion and oxidation conditions. Here are some specific comments and suggestions for improvement:

1. What factors might contribute to the significant weight gain during the air oxidation test in Case II and Case III as compared to Case I?

2. Regarding Figure 1(a), it’s mentioned that the weight gain of the samples increased rapidly with parabolic behavior until 500 h and then slowed down following linear behavior. Can you explain more about the mechanism of this behavior?

3. The formation of Fe-rich oxides is discussed in the context of the autoclave material, but the implications of this finding are not thoroughly explored. How might the presence of Fe-rich oxides influence the long-term integrity of the oxide layer, especially in the context of dry storage?

4. It is suggested to strengthen the article by providing more details about the formation mechanism of ZrO2.

5. “It is anticipated that the presence of bubble would not affect the overall integrity of oxide layer during the long-term storage in dry storage facility” Please explain the practical implications of the observed bubble formation and how it does not affect the integrity of oxide layer during long-term storage.

Comments on the Quality of English Language

Language is ok.

Author Response

See the attached file for the answers to the comments.

Thank you

Reviewer 2 Report

Comments and Suggestions for Authors

The article aims to simulate an accelerated production of a 10 μm thick oxide layer on the Zircaloy-4 cladding in conditions similar to the CANDU spent fuel cladding by combining high-temperature water corrosion and air oxidation. The research subject is quite interesting and significant for the field of spent nuclear fuel storage. The authors should emphasize the originality of the study, compared to other references cited in the text. The current form of the manuscript requires minor revisions before publication in the Materials journal.

Some questions and observations are summarized below:

1.      The authors should carefully proofread and spell check to eliminate grammatical and spelling errors like: line 223 to the dissolve Fe from; line 285: oxide layer was mixture; line 330: are only differ in thickness; line 338: in oxygenated high-temperature water at (different font was detected – similar for line 354-357); line 351: oxide layer is mainly consisted of;

2.      All the acronyms used in the manuscript should be explained in text. (e.g. CANDU)

3.      The authors should mention the provider of the materials involved in the study (e.g. Zircaloy-4).

4.      Regarding the preparation method of the substrate, were the specimens ground or polished to a specific roughness? Does the roughness have an influence on the corrosion behavior of the material? Please explain.

5.      How many specimens were investigated for each case of corrosion test protocol? In Figure 1a is also presented a standard deviation of the weight change value.

6.      An additional XRD analysis including the Zircaloy-4 before the high-temperature oxidation and air oxidation should be included, to emphasize the phase changes during the exposure to the aggressive environment.

Comments on the Quality of English Language

Minor editing of English language required.

Reviewer 3 Report

Comments and Suggestions for Authors

Dear Authors,

In your paper, you describe studies that are not popular, but interesting. The scope of the study is clearly presented and the drawings are of good quality. My comments, as a reviewer, relate to several issues:

Note 1: Before section 2, you must add a section that will be a review of literature, maybe studies of a different scope on the same material (zirconium). This section does not have to be titled: Literature review, but can be background of research (research background). You can move some of the text from the Introduction to a new section.

Note 2: When you add a literature section (note 1), your references will be better (more scientific publications in the paper references).

Note 3: For the Experimental Methods section, you have to write or add a study diagram with 3 cases.

Note 4: results. The formula must be numbered (1). In my opinion, this formula and the text about them must be moved to 2. 3. Analysis methods

Note 5: Line 147 , two ))     see:  In Case II (Figure 1(b))

Note 6: There is no discussion in your paper, it doesn’t have to be, but you can mention in the summary the limitations of your research or the application of your research, or further research.

Best wishes

Reviewer

Round 2

Reviewer 1 Report

Comments and Suggestions for Authors

I have no more comments.